# The impact of the diurnal cycle of the atmospheric boundary layer on physical variables relevant for wind energy applications

Antonia Englberger<sup>1</sup> and Andreas Dörnbrack<sup>1</sup> <sup>1</sup>Institut für Physik der Atmosphäre, DLR Oberpfaffenhofen *Correspondence to:* Antonia Englberger (Antonia.Englberger@dlr.de)

Abstract. This paper provides a quantification of the temporal evolution of physical variables in the atmospheric boundary layer (ABL) relevant for wind energy applications. For this purpose, we use the unique dataset gathered during the BLLAST (Boundary Layer Late Afternoon and Sunset Turbulence) field experiment to validate a large-eddy simulation (LES) model by simulating

- the complete diurnal cycle of the ABL. In this way, this contribution to the special issue of ACP 'The Boundary-Layer Late Afternoon and Sunset Turbulence project' satisfies the purpose of the BLLAST experiment: to provide a dataset for the validation of numerical simulations aiming to study transient BL processes. For wind energy applications, we are investigating the behaviour of different physical parameters which are relevant in the height region where a wind turbine operates.
- This results in a quantification of the diurnal cycle influence on the vertical wind shear, the stratification and the turbulence intensity in the atmosphere. Further, the impact of different heterogeneous surface conditions on shear near the surface layer of the ABL is investigated.

# 1 Introduction

- The ABL is characterised by a diurnal cycle, which is forced by the solar irradiation, heating the
  Earth's surface during daytime and the infrared radiation to space, cooling the Earth's surface during night. It is additionally influenced by mesoscale and synoptic scale external forcings. The diurnal cycle (Stull (1988), p 11, Fig. 1.7) is composed of the stable boundary layer (SBL) at night and the convective boundary layer (CBL) or neutral boundary layer (NBL) during daytime. In between the night and day regimes different transient periods exists which are named morning (MT), afternoon
  (AT) and supplies transition (ET) respectively.
- (AT) and evening transition (ET), respectively.

The diurnal cycle is influenced in detail by the following physical processes (Stull (1988)): During the night, when the surface fluxes are rather low, the BL consists of an SBL capped by a neutrally stratified layer, the residual layer (RL). The RL results from the decay of turbulence of the CBL

of the previous day. The SBL is characterised by a small turbulence intensity, because the eddies

have less energy in comparison to the CBL due to less surface fluxes. An increase of the surface fluxes from its minimum level initiate the onset of the MT. The heating of the surface generates thermals which erode the stable layer from below. During the MT, the turbulent eddies increase in size and strength and start to form a fully convective layer, which continues to grow throughout the

- morning. During this transition period, the mixed layer incorporates the RL from the previous night. After this process called free encroachment (Sorbjan (2004)) has happened, a fully developed CBL evolves from the MT and is present during daytime. In contrast to the SBL, the CBL is characterised by a larger amount of turbulence, which is generated by the domination of buoyancy over shear. The buoyancy results from upward motion of air in thermal plumes initiated by a positive heat flux
- from the surface. The CBL is not only influenced from below, entrainment processes incorporate non-turbulent air from the free atmosphere above. These processes involve fluxes of heat, mass, momentum and moisture across the BL top. The updraughts and downdraughts lead to convective motions in the CBL, which are suppressed by the stratification of the atmosphere in the SBL. When the forcing at the surface declines due to decreasing heat fluxes, the afternoon transition sets in.
- Different definitions of the onset of the AT exist (e.g. Nadeau et al. (2011), Grimsdell and Angevine (2002)). We consider as AT the time from the decrease of the surface sensible heat flux up to the time the ET sets in. The onset of the ET is initiated as the time approaching zero sensible heat flux at the surface (e.g. Grimsdell and Angevine (2002)). During the ET, the decaying CBL merges into the SBL and a RL above the SBL, closing the diurnal cycle.

The behaviour of this diurnal cycle of the ABL has been studied since the 1970s. There are many studies (observational and numerical) regarding the SBL (e.g. Nieuwstadt (1984), Carlson and Stull (1986), Mahrt (1998)) or especially the RL (e.g. Balsley et al. (2008), Wehner et al. (2010)). Also the CBL has been investigated intensively over the last decades with different focuses, e.g. on entrainment (e.g. Sorbjan (1996), Sullivan et al. (1998) and Conzemius and Fedorovich (2007)) and on shear (e.g. Moeng and Sullivan (1994), Fedorovich et al. (2001), Pino et al. (2003) and Pino et al.

(2003)). Deardorff (1974a) and Deardorff (1974b) performed the first LES of a transition process in the ABL. Since then, many LES of the transitional phases have been performed on the MT (e.g. Sorbjan (2007) and Beare (2008)) as well as the AT (e.g. Sorbjan (1997), Sorbjan (1996), Beare et al.
(2006) and Pino et al. (2006)).

Regarding the different performed studies, we address as first research question:

- (i) Is it possible to validate the complete diurnal cycle of the ABL for the BLLAST dataset with our LES model EULAG?
- The reason to ask this question is the known strong influence of the diurnal cycle on the physical quantities in the lowest 200 m of the ABL (Emanuel et al. (2015)). Therefore, a validated numerical method is required to provide reliable profiles during the whole day, especially in the transition

phases. The interaction of the ABL turbulence with wind turbine wakes is only poorly understood due to the complexity of the ABL (Naughton et al. (2011), Emeis (2013), Emeis (2014)). This paper can be considered as an investigation to prepare LESs of wind turbine wakes subject towards realistic

- can be considered as an investigation to prepare LESs of wind turbine wakes subject towards realistic ABL regimes. Therefore, the BLLAST dataset is used to validate our results and to compare them with the accompanying numerical simulations published in ACP (e.g. Blay-Carreras et al. (2014), Nilsson et al. (2015)).
- According to both, experimental studies (e.g. Medici and Alfredsson (2006), Chamorro and Porté-Agel (2009), Zhang et al. (2012)) and numerical simulations (e.g. Troldborg et al. (2007), Wu and Porté-Agel (2012)) of wind turbines, the inflow wind field a wind turbine is exposed to strongly influences the wake structure and the turbine loading, both affecting the power production of a wind turbine. Therefore, the wake structure strongly depends at least on the following three physical pa-
- rameters: the vertical wind shear, the stratification of the atmosphere and the amount of turbulence in the atmosphere. These parameters, however, strongly vary in the different phases of the diurnal cycle (Stull (1988)). There are experimental studies considering different atmospheric stratifications (NBL, SBL, CBL) (e.g. Medici and Alfredsson (2006), Chamorro and Porté-Agel (2010), Zhang et al. (2012), Tian et al. (2013), Zhang et al. (2013)). In most of the numerical simulations of an
- individual wind turbine, however, an NBL is assumed (e.g. Wu and Porté-Agel (2011), Porté-Agel et al. (Chapel Hill, 2010), Naughton et al. (2011), Englberger and Dörnbrack (2015)). There are approaches considering the SBL (Aitken et al. (2014)) or the CBL (Mirocha et al. (2014)) in a one-way nested Weather Research and Forecasting (WRF)-LES simulation. Some recent LES studies start to investigate the impact of an SBL on the wake (Bhaganagar and Debnath (2014) and Bhaganagar and
- Debnath (2015)). However, to our knowledge, no study has been performed so far investigating the influence of all the phases of the diurnal cycle, including the CBL and the transitions. Therefore, the second research question is:
  - (ii) Which impact have the individual phases of the diurnal cycle on the physical variables relevant for wind energy applications?
- This study is performed as an intermediate step towards an investigation placing an emphasis on the interaction of the different atmospheric stratifications with the flow field behind a wind turbine. Such a study is necessary, as the effect of different atmospheric conditions on the wake is varying over the course of the diurnal cycle and the propagation and dissipation of the wake is not well understood up to now (Emanuel et al. (2015)).

Most of the performed LES simulations on the characteristics of the BL, mentioned above, prescribe homogeneous surface conditions. However, the Earth's surface is not homogeneous. It is strongly affected by different land use, buildings, and so on. Therefore, considering heterogeneous surface conditions will especially improve the turbulence structure close to the ground. According to e.g.

110

- Stull (1988), Moeng and Sullivan (1994) or Beare (2008), the diurnal cycle is affected by different stratifications of the atmosphere which have a large impact on the amount of turbulent kinetic energy (TKE) in the BL. At daytime, the TKE budget is dominated by a buoyancy driven structure, whereas at night shear dominates. The buoyancy results from the applied surface fluxes. The shear is influenced by the surface heterogeneity. For example, according to Dörnbrack and Schumann (1993),
- for ratios of the friction velocity  $u_*$  to the convective velocity scale  $w_*$  larger than 0.35 shear becomes important over heterogeneous surfaces. The main effect of a heterogeneous surface on shear arises close to the ground, in the area relevant for wind energy applications. This results in our third research question:
  - (iii) Which impact has the heterogeneity of the surface on shear, contributing to the TKE budget, in a region relevant for wind energy applications?

The results of this study pave the groundwork for future work in analysing the interaction of a wind turbine in a turbulent boundary layer under different atmospheric stratifications and a realistic heterogeneous surface.

- The outline of the paper is as follows. The numerical model, the diurnal cycle representation, the applied external forcings and the heterogeneous surface method are described in chapter 2. The observations, the numerical experiment and the validation of the complete diurnal cycle for the BLLAST dataset with our LES model EULAG are presented in chapter 3. Chapter 4 investigates the impact of the individual phases of the diurnal cycle on different physical variables relevant for
- wind energy applications. Chapter 5 explores the impact the surface heterogeneity on shear. The discussion of the paper is presented in chapter 6 and a summary is given in chapter 7.

## 2 Framework

# 2.1 Numerical model

The ABL is simulated with our numerical model EULAG, which is at least of second order accuracy
in time and space and well suited for massively-parallel computations (Prusa et al. (2008)). It can be run parallel up to a domain decomposition in three dimensions. The name EULAG refers to the ability of solving the equations of motions either in an EUlerian (flux form) (Smolarkiewicz and Margolin (1993)) or in a semi-LAGrangian (advective form) (Smolarkiewicz and Pudykiewicz (1992)) mode. A comprehensive description and discussion of EULAG can be found in Smolarkiewicz and Margolin (1998) and Prusa et al. (2008).

For the following simulations, the Boussinesq equations for a flow with constant density  $\rho_0 = 1.1225 \text{ kg m}^{-3}$  are used and the environmental state ( $\Theta_0$ ) is described by an isotherm with  $\Theta_0 =$ 

301 K. The governing equations are written for velocity components in Cartesian coordinates in Equations (1), (2) and (3) (Lipps and Hemler (1982)).

$$\frac{d\mathbf{v}}{dt} = -G\boldsymbol{\nabla}\left(\frac{p'}{\rho_0}\right) + \mathbf{g}\frac{\Theta'}{\Theta_0} + \boldsymbol{\mathcal{V}} + \mathbf{M} + \mathbf{F}_v - \alpha_m \mathbf{v} \tag{1}$$

$$\frac{d\Theta'}{dt} = \mathcal{H} + F_{\Theta} - \alpha_h \Theta' \tag{2}$$

$$\boldsymbol{\nabla} \cdot (\rho_0 \mathbf{v}) = 0 \tag{3}$$

Here,  $\frac{d}{dt}$ ,  $\nabla$  and  $\nabla$  · represent the total derivative, the gradient and the divergence. v denotes the 140 physical velocity vector, p' the pressure perturbation with respect to the environmental state, **g** the gravitational acceleration and  $\rho_0$  the density of the fluid.  $\Theta'$  is the deviation of the potential temperature  $\Theta$  from the environmental state of the atmosphere  $\Theta_0$ . G represents the Jacobian of the transformation which result from the general, time-dependent coordinate transformation (Smolarkiewicz and Prusa (2005), Prusa et al. (2008)). The subgrid scale terms  $\mathcal{V}$  and  $\mathcal{H}$  symbolise viscous dissi-

- pation of momentum and diffusion of heat. **M** denotes the inertial forces of coordinate-dependent metric accelerations.  $\mathbf{F}_v$  and  $\mathbf{F}_{\Theta}$  are the additional forcing applied on the wind and the potential temperature perturbation. The relaxation terms  $\alpha_m$  and  $\alpha_h$  are used to represent a heterogeneous surface. The surface obstacle elements are included via the immersed boundary method (Smolarkiewicz et al. (2007)). This method mimics the presence of solid structures and internal boundaries by applying
- the fictitious body forces  $-\alpha_m \mathbf{v}$  in Equation (1) and  $-\alpha_h \Theta'$  in Equation (2). In the fluid away from the solid boundaries  $\alpha_m$  and  $\alpha_h$  are both zero, whereas they are approaching  $\frac{1}{2}\Delta t$  within the solid assuring the velocity to approach zero.  $\Delta t$  represents the time step.
- In general, EULAG owes its versatility to a unique design that combines a rigorous theoretical formulation in generalized curvilinear coordinates (Smolarkiewicz and Prusa (2005)) with nonoscillatory forward-in-time (NFT) differencing for fluids built on the multidimensional positive definite advection transport algorithm (MPDATA), which is based on the convexity of upwind advection (Smolarkiewicz and Margolin (1998); Prusa et al. (2008)) and a robust, exact-projection type, elliptic Krylov solver (Prusa et al. (2008)). The flow solver has been applied to a wider range of scales
- simulating various problems like turbulence (e.g. Smolarkiewicz and Prusa), flow past complex or moving boundaries (e.g. Kühnlein et al. (2012)), gravity waves (e.g. Doyle et al. (2011)) or even solar convection (e.g. Smolarkiewicz and Charbonneau (2013)).

## 2.2 Diurnal cycle representation

The simulations of an SBL and a CBL have to meet different requirements. For the simulation of an SBL, a fine spatial resolution is needed to represent the small size eddies and a small domain is sufficient. A CBL simulation, however, requires a large enough domain to capture the convective

processes whereas a coarse computational grid is satisfying. The requirements simplest approach of combining these results in a large domain with a rather fine resolution. However, this is a computationally very expensive approach. Therefore, for the transition from an SBL towards a CBL, the

- domain expansion method from Beare (2008) is applied. For the transition back from the coarse to the fine resolution the domain size cannot simply be decreased again, as the same physical situation should be represented until the end of the simulation. Therefore, the domain size is kept constant and the resolution is decreased by performing an interpolation procedure. Both transition methods are conducted in two steps separated by one hour of physical time to limit numerical instabilities. We,
- hereafter, refer to them as MT method and ET method. In Table 1 the spatial horizontal resolutions and domain sizes for the corresponding atmospheric regimes are listed.

## 2.3 External forcings

The diurnal cycle is controlled by the heating or cooling of the Earth's surface, described by surface fluxes. To simulate the diurnal cycle realistically, additional external forcings as mesoscale and syn-

- optic scale processes have to be modelled and integrated into the LES, which is a computationally expensive process. The alternative we choose for this paper is to incorporate external forcings via physical parameterisations in the LES. In this investigation we include subsidence (S) and radiative cooling as large-scale processes.
- The surface fluxes prescribe the diurnal heating cycle, with a cooling at night and a warming during daytime. In general, a sinusoidal like function is assumed with the maximum value occurring at the time around noon corresponding to the maximum in solar irradiation (e.g. Sorbjan (2007), Beare (2008)). The solar or infrared irradiation value divided by  $\rho_0$  and the specific heat capacity at constant pressure corresponds to the surface fluxes, which are contributing to Equation (2). The surface
- fluxes in the lowest level are controlled by the subgrid-scale model. In all other levels we vertically distribute the surface fluxes with height with an e-folding scale of 300 m via  $SF \cdot exp(-\frac{z}{300m})$ .

The subsidence process is a large-scale process, representing the main physical mechanism responsible for the decent of the CBL height. According to Mazzitelli et al. (2014), subsidence does not only affects the height of the CBL it also influences the turbulent fluctuations within the CBL. The subsidence process is important in simulations aiming at a realistic simulation of the ABL to match actual measurements of temperature and turbulence in the atmosphere. In general, subsidence acts on the zonal and meridional wind components u and v, as well as on the potential temperature Θ. It can be modelled by the product of a subsidence velocity w<sub>sub</sub> and the vertical derivative of the wind components or temperature, according to Equation (4). Here, ξ equals u and v in Equation (1) and

 $\Theta'$  in Equation (2).

$$S_{i,j,k}^{\xi} = w_{sub} \frac{d\xi}{dz} = w_{sub} \frac{\xi_{i,j,k+1} - \xi_{i,j,k}}{z_{i,j,k+1} - z_{i,j,k}}$$
(4)

The indices of the grid points are denoted by i = 1 ... n, j = 1 ... m, and k = 1 ... l in the x, y and z directions, respectively. The velocity w<sub>sub</sub> determines the strength of subsidence, with a larger
subsidence velocity resulting in a lower CBL height. In the following simulations, the subsidence velocity is constant with height with S<sub>i,j,k</sub> acting in all levels and grid points besides at the ground and the top. The parameterisation acts as S<sup>u</sup>=F<sub>u</sub> and S<sup>v</sup>=F<sub>v</sub> in Equation (1), whereas S<sup>Θ</sup> is another contribution of F<sub>Θ</sub> in Equation (2).

- There are different parameterisations of the subsidence process. An increase of  $w_{sub}$  from zero at the ground up to a maximum value of the order of  $\mathcal{O}(10 \text{ mm s}^{-1})$  at the top of the BL and a constant value of this maximum in the free atmosphere is typically used in the literature (e.g. Blay-Carreras et al. (2014)). We use a constant subsidence velocity to result in a larger effect of subsidence in the lower levels. This is motivated as we have no knowledge about individual processes resulting in a
- height dependent subsidence velocity. A parameterisation is considered to be reasonable if it reaches the maximum of subsidence at the top of the boundary layer in order to control the height of the BL top. Equation (4) verifies this requirement because the vertical gradient  $\frac{d\xi}{dz}$  maximises at the transition from the BL to the inversion layer. This dependence controls the height of the BL, whereas the magnitude of the subsidence velocity influences the strength of this change.

The applied subsidence on the potential temperature in Equation (2) does not only result in a decrease of the boundary layer height, it also warms the boundary layer air. Therefore, radiative cooling has to be considered as another large-scale process in order to compensate the additional warming making the simulations conform with the measurements. In our simulations radiative cooling is con-

stant with height and acts in all levels and grid points besides at the ground and the top. It is also applied on  $F_{\Theta}$  in Equation (2), resulting in  $F_{\Theta} = SF \cdot exp(-\frac{z}{300m}) + S^{\Theta} + \frac{\partial \Theta}{\partial t}\Big|_{rad}$  for z > 0.

## 2.4 Surface heterogeneity

The main simulation is performed with a homogeneous surface, a drag coefficient of 0.1 and a horizontally homogeneous momentum flux prescribed over the complete domain. In three additional simulations, a heterogeneous surface is applied to analyse the impact of the surface heterogeneity on the ABL. The main surface characteristics are listed in Table 2. In simulation B, cubic obstacles with a size of 100 m × 100 m × 5 m, separated by 100 m from each other in zonal and meridional direction, are implemented via the immersed boundary method in Equations (1) and (2). These obstacles are acting for example as individual patches of different land use or buildings. To test the sensitivity

of the obstacles, simulation B is repeated with two times the height of the obstacles in simulation C

 $(100 \text{ m} \times 100 \text{ m} \times 10 \text{ m})$ . In simulation D, the area covered by the obstacles is modified, whereas the size of the obstacles stays the same as in simulation B. Here, the obstacles are separated by 200 m from each other, leading to one fourth of the area covered by cubes in comparison to simulation B. In all modified simulations (B to D), local variations regarding the momentum flux are prevalent.

## 240 3 Diurnal cycle validation

## 3.1 Observations

The BLLAST field campaign took place from 14 June to 8 July 2011 in southern France at Lannemezan, an area of complex terrain a few kilometre from the Pyrenean foothills. The heterogeneous surface in the plateau is covered by grass, meadows, crops and forest. Measurements from towers,

radiosondes, airplanes and ground-based remote sensing are recorded during the whole campaign, with twelve intensive observing periods (IOPs), characterised by various meteorological conditions. A detailed description can be found in Lothon et al. (2014).

We use the IOP of the BLLAST campaign from the 1st of July 2011 (Lothon et al. (2014)) for the
validation of our method of the simulation of a diurnal cycle with the LES model EULAG. We chose
the 1st of July because it is characterised by anticyclonic conditions minimising large-scale forcings
except subsidence and radiative cooling. The lower troposphere is governed by a mountain-plain
circulation along the valley with almost no clouds prevalent. The data of four radiosonde launches
at 0000 UTC, 1100 UTC, 1658 UTC and 2254 UTC are available for the 1st of July 2011.

#### 255 3.2 Numerical experiment

260

Our diurnal cycle simulation is initialised with the potential temperature data from the 0000 UTC radiosonde launching. The development of the meteorological fields with time are compared to and validated against the other three measurements. Our numerical simulation runs for 23 h, starting with the SBL, including the MT, the CBL phase, the AT, the ET and finishing with the SBL, modelling a complete diurnal cycle of the ABL.

By performing the diurnal cycle method for a simulation of the complete day, we start the SBL simulation with a domain size of 793.75 m x 793.75 m x 3 km, resulting in a domain size of 3.175 km x 3.175 km x 3 km after the MT up to the end of the simulation, following Table 1. The vertical
resolution is set to 5 m in the first 200 m, increased to 10 m up to 800 m and 20 m to the domain top at 3 km.

The flux measurements are taken from a 60 m tower with values at 30 m, 45 m and 60 m altitude. The sensible heat flux at the surface is calculated via an e-folding scale of 300 m and the measure-