# Peer review of "The impact of the diurnal cycle of the atmospheric boundary layer on physical variables relevant for wind energy applications"

_Atmospheric Chemistry and Physics, 2015_

## Referee Comment (RC1) · Anonymous Referee #1 · 29 Feb 2016

Review of "The impact of the diurnal cycle of the atmospheric boundary layer on physical variables relevant for wind energy applications" by Englberger and Dörnbrack.

The authors present large-eddy simulations of a diurnal cycle loosely based on thermodynamic observations from the BLLAST experiment. They compare their simulations, which are tuned with subsidence and radiative cooling, to temperature profiles from BLLAST, but they increase the winds in their simulations by a factor of 3 from those in the observations. The authors do not acknowledge previous work on LES of the diurnal cycle (Kumar et al. 2006; Basu et al. 2008) and claim to be performing the first LES

of the diurnal cycle (lines 85-86). Details of the simulations, including presentation of wind and turbulence profiles, as well as components of the TKE budgets are presented. Winds and turbulence in the lowest 200 m are discussed in detail "to expose the impact of the individual phases of the diurnal cycle on these physical variables which are relevant for wind energy applications", but novel insights are not provided and the authors fail to refer to previous simulations or observations which have explored the impact of the diurnal cycle on wind-energy-relevant quantities. Although there may be novel contributions in this work, the present manuscript does not highlight such contributions in a satisfactory way. Several concerns are outlined below, along with suggestions that could help the authors refocus a revised manuscript.

Part of the confusion in the presentation may be due to a lack of focus because the simulations are not placed in a proper context: instead of highlighting any novel aspects of these simulations, the authors instead focus on an interesting challenge that is unrelated to the simulations discussed in this manuscript. Specifically, much space in the introduction is devoted to a summary of large eddy simulations of wind turbine wakes (lines 63-95) although the present study does not include wind turbines. If this work is an intermediate step toward LES of wind turbine wakes, the present study should still be unique and novel enough to stand on its own. The authors could focus on the diurnal cycle of the ABL with their LES, providing more details on some of their technical approaches (nesting, immersed boundary method for canopies, subgridscale turbulence modeling) – these are important aspects of their approach that are neglected in the discussion. Further, previous contributions that have already carried out LES of a diurnal cycle are omitted from the literature review (Kumar et al. 2006; Basu et al. 2008). The authors should review these papers and consider how the present work provides a unique contribution.

Further, the correspondence of these simulations to the BLLAST observations is questionable. The authors compare their potential temperature profiles to the observed potential temperature profiles at only three points in the diurnal cycle (one profile is

used for initialization, three for evaluation). No data other than soundings is used for evaluation although BLLAST included considerable instrumental deployments. They use two tuning parameters (subsidence and radiative cooling) to achieve approximate agreement with the profiles (but, as noted below, the authors do not refer to previous work on LES with subsidence). How should a reader develop confidence in the selection of subsidence rate and cooling rate? Are there any observations that support these choices of subsidence or radiative cooling? Second, the authors modify the winds in their simulations substantially from the observed 3 m s-1 to 10 m s-1 (a factor of three!) but still suggest that their simulations compare well to the BLLAST observations. It would be a cleaner comparison to first match both the winds and the thermodynamics (so that they can validate simulations with observed fluxes, aircraft data, etc.). Later, once the reader trusts the simulations, the authors could increase the winds if necessary. As the simulations stand right now, they are not really based on any observations with so many tuning parameters and vastly different winds. If the authors really require winds of 10 m s-1, they should find another experiment (CASES-99? Numerous studies from Cabauw?) that can provide adequate data for validation.

Finally, the authors do carry out a small ensemble of simulations with varying canopies and obstacles in the flow, emphasizing the novelty of surface heterogeneity in LES of the ABL. Again, they have not reviewed previous work in this area (Belcher et al. 2003; Kang et al. 2012, among others), nor have they supplied sufficient details on their implementation of the immersed boundary method for readers to understand if this is a novel contribution. Nevertheless, there could be novel aspects to this part of their study that would justify a publication on their LES.

Please note that although I am recommending rejection of the manuscript in its current form, I do encourage the authors to think carefully about how to reframe this work to identify and emphasize novel contributions.

Numerous comments regarding references, clarify, grammar, and scientific concerns are found below, listed by line number.

27: "its minimum level" should be "their minimum level" (referring to surface fluxes)

35: should read "The CBL is not only influenced from below, but also from above, as entrainment processes incorporate. . ."

37: updraft and downdraft instead of "updraught" etc.

40: why are these citations not in chronological order?

46-55: these lists of citations are not put into context for the reader

58: this research question is not particularly innovative as LES of the diurnal cycle has been carried out (Kumar et al. 2006; Basu et al. 2008) (and these papers should be cited)

61: Why is a recent paper on wind energy research (Emanuel et al. 2015) referred to for discussion of the diurnal cycle of the ABL when many other older and more detailed studies of the diurnal cycle of the ABL for wind energy applications could be considered? should cite (Barthelmie et al. 1996; Walter et al. 2009; Storm and Basu 2010; Rhodes and Lundquist 2013, among others)

63-64: several papers have investigated the impact of atmospheric stability on wind turbine wakes (Aitken et al. 2014; Bhaganagar and Debnath 2015; Mirocha et al. 2015, for example)

73: wind turbine loading should refer to (Sathe et al. 2013)

79: the comment that "most papers assume NBL" is not consistent with the subsequent citations focusing on the SBL and CBL

87: The discussion states that no other papers have explored how the complete diurnal cycle influences wind turbine wakes, but that is not the point of this present manuscript. This paper focuses on parameters relevant to wind energy applications, which is different from wind turbine wakes. It feels like the authors are trying to define a specific niche, not very accurately, and without correspondence to the goal of the present paper. The authors should think carefully about the specific novel contributions of the specific work presented here, and emphasize those contributions, not necessarily the larger project.

144: the subgridscale model is very important but is not discussed in any detail here. Is it a TKE-based model? More details are necessary.

167: "satisfactory" rather than "satisfying"

175: Please explain that "MT" stands for "morning transition" and similarly for "ET"

194: "descent" rather than "decent"; check grammar in this sentence,

194: discussion of subsidence is interesting but very focused on recent work. Please also refer to previous work on subsidence such as (Mirocha and Kosović 2009). In fact, given that subtle changes in the specification of subsidence can significantly change surface fluxes and boundary-layer height, how can the authors justify their choice of w_sub (and the resulting choice for radiative cooling?)

233: more explanation or references are required for the "immersed boundary method" – many choices for how to implement the immersed boundary method are possible.

258: validation of LES is carried out only against radiosondes? What about flux measurements and the multiple other measurements possible during BLLAST?

Figure 1: difficult to interpret for red-green color-blind readers. Can the lines be thicker and labelled directly instead of relying on the legend? Are axis labels consistent with ACP requirements?

271: should be "W m-2" rather than "W m2"

283: it appears that the authors are attempting to match the thermodynamic aspects of the observations but for a difference range of wind speeds (10 m/s instead of the observed 3 m/s). Wouldn't the thermodynamic profiles have changed with different winds? The utility of the comparison to observations seems lacking, and it is not clear

that "validation" describes what is happening here.

300-315: this "validation" is really a qualitative comparison. Can quantitative metrics be calculated?

311-314: speculation on what tweaks could improve model performance are not appropriate: either make the changes and evaluate the simulations or avoid commenting on possibilities.

316-319: sentence unclear

334-349: these observations are not particularly novel or innovative for LES of a CBL.

356: Please refer to wind energy industry references for assertions of "typical" parameters. The Global Wind Energy Council has appropriate statistics.

366: The variability of the wind profile between daytime and nighttime is well known and documented in the literature already (Barthelmie et al. 1996; Walter et al. 2009; Rhodes and Lundquist 2013)

370-417: The discussion of the TKE budget does not introduce novel insights compared to previous studies such as (Beare et al. 2006)

440: The main conclusion seems to be the effect of obstacles on shear. Again, the novelty of this result is questionable (see previous work, such as (Belcher et al. 2003))

Aitken ML, Kosović B, Mirocha JD, Lundquist JK (2014) Large eddy simulation of wind turbine wake dynamics in the stable boundary layer using the Weather Research and Forecasting Model. J Renew Sustain Energy 6:033137. doi: 10.1063/1.4885111

Barthelmie RJ, Grisogono B, Pryor SC (1996) Observations and simulations of diurnal cycles of near-surface wind speeds over land and sea. J Geophys Res Atmospheres 101:21327–21337. doi: 10.1029/96JD01520

Basu S, Vinuesa J-F, Swift A (2008) Dynamic LES Modeling of a Diurnal Cycle. J Appl

[Figure]

Meteorol Climatol 47:1156–1174. doi: 10.1175/2007JAMC1677.1

Beare RJ, Macvean MK, Holtslag AAM, et al (2006) An Intercomparison of Large-Eddy Simulations of the Stable Boundary Layer. Bound-Layer Meteorol 118:247–272. doi: 10.1007/s10546-004-2820-6

Belcher SE, Jerram N, Hunt JCR (2003) Adjustment of a turbulent boundary layer to a canopy of roughness elements. J Fluid Mech 488:369–398. doi: 10.1017/S0022112003005019

Bhaganagar K, Debnath M (2015) The effects of mean atmospheric forcings of the stable atmospheric boundary layer on wind turbine wake. J Renew Sustain Energy 7:013124. doi: 10.1063/1.4907687

Kang S-L, Lenschow D, Sullivan P (2012) Effects of Mesoscale Surface Thermal Heterogeneity on Low-Level Horizontal Wind Speeds. Bound-Layer Meteorol 143:409–432. doi: 10.1007/s10546-011-9691-4

Kumar V, Kleissl J, Meneveau C, Parlange MB (2006) Large-eddy simulation of a diurnal cycle of the atmospheric boundary layer: Atmospheric stability and scaling issues. Water Resour Res 42:W06D09. doi: 10.1029/2005WR004651

Mirocha JD, Kosović B (2009) A Large-Eddy Simulation Study of the Influence of Subsidence on the Stably Stratified Atmospheric Boundary Layer. Bound-Layer Meteorol 134:1–21. doi: 10.1007/s10546-009-9449-4

Mirocha JD, Rajewski DA, Marjanovic N, et al (2015) Investigating wind turbine impacts on near-wake flow using profiling lidar data and large-eddy simulations with an actuator disk model. J Renew Sustain Energy 7:043143. doi: 10.1063/1.4928873

Rhodes ME, Lundquist JK (2013) The Effect of Wind-Turbine Wakes on Summertime US Midwest Atmospheric Wind Profiles as Observed with Ground-Based Doppler Lidar. Bound-Layer Meteorol 149:85–103. doi: 10.1007/s10546-013-9834-x

Sathe A, Mann J, Barlas T, et al (2013) Influence of atmospheric stability on wind turbine loads: Atmospheric stability and loads. Wind Energy 16:1013–1032. doi: 10.1002/we.1528

Storm B, Basu S (2010) The WRF Model Forecast-Derived Low-Level Wind Shear Climatology over the United States Great Plains. Energies 3:258–276. doi: 10.3390/en3020258

Walter K, Weiss CC, Swift AHP, et al (2009) Speed and Direction Shear in the Stable Nocturnal Boundary Layer. J Sol Energy Eng 131:011013. doi: 10.1115/1.3035818

---

## Referee Comment (RC2) · Anonymous Referee #2 · 5 Mar 2016

General comments:

The paper discusses an LES study of a full diurnal cycle of the ABL that is validated with data from the BLLAST Field program. The study is an intermediate step toward a full LES model that incorporates wind turbine modeling. I have three general concerns regarding the manuscript. The first concern is that the paper does not appear to be sufficiently novel to stand as an independent publication. The contributions of the paper, in their current state, seem more appropriate as a chapter of a larger study. This is due to much of the introduction discussing turbine modeling, something that is not actually

modeled in the paper. The introduction should put less emphasis on turbine modeling and more emphasis on literature relevant to modeling the full diurnal cycle with realistic boundary conditions. A quick Google search reveals that a number of previous LES publications have studied the full diurnal cycle. What is unique and novel about your approach?

Second, the validation with the BLLAST dataset is unconvincing for two reasons. First, the model was run with winds at 10 m s-1 while the observations showed that the wind speed was closer to 3 m s-1. A convincing validation should match wind speeds. Second, the model is only validated with three potential temperature profiles and the comparison is largely qualitative. I'd like to see a quantification of the errors as well as validation with additional observations from the BLLAST campaign. The discussion of the results then shows that the LES model is able to capture the expected structure of the ABL. The discussion fails to highlight what is novel about these results compared with previous LES studies of the ABL

Finally, the grammar, sentence structure and general readability of the manuscript need to be improved. I encourage the authors to carefully edit the manuscript and highlight what is unique about their approach/results and resubmit.

Specific Comments

Line 5: "In this way, this contribution to the special issue of ACP 'The Boundary-Layer Late Afternoon and Sunset Turbulence project' satisfies the purpose of the BLLAST experiment: to provide a dataset for the validation of numerical simulations aiming to study transient BL processes" – This is not necessary to include in the abstract, include major conclusions instead.

Line 15 – It may not be necessary to include so much detail regarding the ABL in ACP. Is it possible to include the ABL schematic from Stull?

Line 34 – "initiated by a positive heat flux" – Use "upward" instead of "positive"

Line 73 – "the inflow wind field a wind turbine is exposed to strongly influences the wake structure and the turbine loading, both affecting the power production of a wind turbine" – Confusing sentence structure. Consider rewording

Line 83 – "Which impact have the individual phases of the diurnal cycle on the physical variables relevant for wind energy applications?" – Confusing, please reword

Line 98 – "Most of the performed LES simulations on the characteristics of the BL, mentioned above, prescribe homogeneous surface conditions. However, the Earth's surface is not homogeneous. It is strongly affected by different land use, buildings, and so on. Therefore, considering heterogeneous surface conditions will especially improve the turbulence structure close to the ground" – Without validation or a reference, this is a non-sequitur conclusion. Just because "realistic" surface conditions are implemented, it does not guarantee that the model will more accurately capture the near-surface turbulence structure. Please include a reference to where this has been validated or show this with your data.

Line 258 – "validated against the other three measurements" – Name the other three measurements Figure 1 Caption – "The initial starting profile is also plotted for 0000 UTC" – That's the dotted line?

Line 309 – "At 2300 UTC, there is a difference prevalent between the measurement and the LES result in the lowest levels." – It'd be nice to see a bit more discussion on this since it's the only simulated profile of the SBL

---

## Author Comment (AC1) · 28 Aug 2016

Manuscript prepared for J. Name
with version 2015/09/17 7.94 Copernicus papers of the LaTeX class copernicus.cls.
Date: 28 August 2016

**Comments of Anonymous Referee 1 on 'The impact of the diurnal cycle of the atmospheric boundary layer on physical variables relevant for wind energy applications' by A. Englberger and A. Dörnbrack**

Antonia Englberger and Andreas Dörnbrack

Antonia.Englberger@dlr.de

First of all, we would like to acknowledge the comments. They were very helpful for improving the manuscript. Below the general comments and the specific comments made by referee 1 are discussed in detail.

Additionally, we changed the title from 'The impact of the diurnal cycle of the atmospheric boundary layer on physical variables relevant for wind energy applications' to 'Impact of the diurnal cycle of the atmospheric boundary layer on wind turbine wakes: A numerical modelling study'.

**General comments**

Referee     The authors present large-eddy simulations of a diurnal cycle loosely based on thermodynamic observations from the BLLAST experiment. They compare their simulations, which are tuned with subsidence and radiative cooling, to temperature profiles from BLLAST, but they increase the winds in their simulations by a factor of 3 from those in the observations. The authors do not acknowledge previous work on LES of the diurnal cycle (Kumar et al. 2006; Basu et al. 2008) and claim to be performing the first LES of the diurnal cycle (lines 85-86)[1.1]. Details of the simulations, including presentation of wind and turbulence profiles, as well as components of the TKE budgets are presented. Winds and turbulence in the lowest 200 m are discussed in detail "to expose the impact of the individual phases of the diurnal cycle on these physical variables which are relevant for wind energy applications", but novel insights are not provided and the authors fail to refer to previous simulations or observations which have explored the impact of the diurnal cycle on wind-energy-relevant quantities[1.2]. Although there may be novel contributions in this work, the present manuscript does not highlight such contributions in a satisfactory way. Several concerns are outlined below, along with suggestions that could help the authors refocus a revised manuscript.

Author     [1.1] There is a misunderstanding. lines 85-86 do not refer to the first large-eddy simulation (LES) of the complete diurnal cycles. We refer it to the investigation of atmospheric variables

relevant in wind energy research not only for stable boundary layer (SBL) and convective boundary layer (CBL), but also for morning transition (MT) and evening transition (ET). In the new version we try to make it more clear with: 'To our knowledge, this is the first study which investigates the influence of a full diurnal cycle, including the ET and the MT, on the wake of a single wind turbine (WT).'

[1.2] We added the references Aitken et al. (2014), Abkar and Porté-Agel (2014), Bhaganagar and Debnath (2014, 2015) and Abkar et al. (2016) to section 4 in the following two paragraphs:

'The diurnal cycle impact on the streamwise wind speed at a certain height is rather small, whereas the variation between top tip, hub height and bottom tip increases during the night (Fig. 5(a)). This results in an influence of the different regimes on the vertical wind shear in Fig. 5(c), similar to investigations of Abkar and Porté-Agel, 2014, Fig. 2a and Abkar et al., 2016, Fig. 5a. Specifically, in the CBL and during the ET the vertical wind shear is rather small, whereas in the SBL and during MT it is very pronounced. A supergeostrophic situation prevails during the MT near hub height corresponding to a low-level jet (LLJ) with a change in wind shear from a positive value below to a negative value above. A supergeostrophic situation also exists in the SBL simulation of Aitken et al., 2014, Fig. 4, Bhaganagar and Debnath, 2014, Fig. 1a and Bhaganagar and Debnath, 2015, Fig. 1. In our simulation, the LLJ is not yet prevalent in the SBL, only a positive wind shear exists between bottom tip and hub height, because the onset time as well as the height of the LLJ depend on the amount of infrared irradiation at night and on the atmospheric situation of the previous day (Bhaganagar and Debnath, 2014, 2015).'

'The diurnal cycle has a large impact on the streamwise turbulent intensity with a maximum during the day and a minimum during the night (Fig. 5(b)), because the negative buoyancy damps the turbulence at night (Fig. 4(c)).The impact of the stratification on the vertical profiles in Fig. 5(d) results in much larger values for the CBL and during the ET and only small values for the SBL and during the MT. The diurnal behaviour, the order of magnitude and the vertical profiles in Fig. 5(b) and (d) are in agreement with investigations of Abkar and Porté-Agel, 2014, Fig. 2d and Abkar et al., 2016, Fig. 3f. Further, the turbulent intensity in Fig. 5(b) and (d) is slightly lower on day 2. This is related to the more stable atmospheric stratification at night 1 in comparison to the initialisation night zero in Fig. 4(b).'

Referee    Part of the confusion in the presentation may be due to a lack of focus because the simulations are not placed in a proper context: instead of highlighting any novel aspects of these simulations, the authors instead focus on an interesting challenge that is unrelated to the simulations discussed in this manuscript. Specifically, much space in the introduction is devoted to a summary of large eddy simulations of wind turbine wakes (lines 63-95) although the present study does not include wind turbines[2.1]. If this work is an intermediate step toward LES of wind

turbine wakes, the present study should still be unique and novel enough to stand on its own. The authors could focus on the diurnal cycle of the atmospheric boundary layer (ABL) with their LES, providing more details on some of their technical approaches (nesting[2.2], immersed boundary method for canopies[2.3], subgridscale turbulence modelling[2.4]) – these are important aspects of their approach that are neglected in the discussion. Further, previous contributions that have already carried out LES of a diurnal cycle are omitted from the literature review (Kumar et al. 2006; Basu et al. 2008)[2.5]. The authors should review these papers and consider how the present work provides a unique contribution.[2.6]

Author [2.1] The introduction still includes a WT discussion, however, this is now justified as we incorporate WT modelling for the different regimes (CBL, ET, SBL, MT) of two days. (Explained in a part of the abstract of the new manuscript:) 'These different characteristics of the atmospheric boundary layer are well-suited for studying the interaction with a wind turbine wake, by applying real-time turbulent inflow data from the idealised atmospheric boundary layer simulation with periodic horizontal boundary conditions to the wind turbine simulation with open horizontal boundary conditions. The resulting wake is strongly influenced by the stability of the atmosphere and recovers more rapidly under convective conditions during the day, compared to the night. The wake characteristics of the transitional periods are influenced by the flow regime prior to the transition.' (More detailed answer in [2.6].)

[2.2] We provide more details about nesting: 'An SBL simulation requires a fine spatial resolution to represent the small size eddies and a small computational domain is sufficient. A CBL simulation requires a large domain to capture the energy-containing thermals and a coarse computational grid is sufficient. Therefore, we initialize an SBL with a horizontal resolution of 6.25 m on 128 x 128 grid points with periodic horizontal boundaries. For the transition from an SBL towards a CBL we apply the domain expansion method from Beare (2008), resulting in a horizontal resolution of 25 m on 128 x 128 grid points in the CBL. For the transition back from the coarse to the fine resolution the domain size is kept constant and the horizontal resolution is decreased to 6.25 m by performing an interpolation procedure, resulting in 512 x 512 horizontal grid points. Both transition methods are conducted in two steps each separated by one hour of physical time to limit numerical instabilities.' instead of lines 164-174.

[2.3] We eliminate the immersed boundary method as simulations with heterogeneous surface are no longer discussed in this manuscript, because we refocus on WT simulations. (More detailed answer in [4.1].)

[2.4] We added information on the subgidscale turbulence model 'All following simulations are performed with a turbulent kinetic energy (TKE) closure.', and added the reference of Smolarkiewicz and Margolin (1998) to Prusa et al. (2008) in line 125.

[2.5] We included previous work on LES of the diurnal cycle of Kumar et al. (2006) and Basu et al. (2008) in the new version of the manuscript in the following paragraph: 'The diurnal

cycle of the ABL has been studied since the 1970s. There are many observational and numerical studies regarding the SBL (Nieuwstadt, 1984; Carlson and Stull, 1986; Mahrt, 1998) or especially the residual layer (RL) (Balsley et al., 2008; Wehner et al., 2010). The CBL has also been investigated intensively with different focuses, e.g. on coherent structures (Schmidt and Schumann, 1989), on entrainment (Sorbjan, 1996; Sullivan et al., 1998; Conzemius and Fedorovich, 2007) and on shear (Moeng and Sullivan, 1994; Fedorovich et al., 2001; Pino et al., 2003). The first LES of a transition process in the ABL was performed by Deardorff (1974a) and Deardorff (1974b). Since then, many simulations considering the transitional phases have been performed on the MT (Sorbjan, 2007; Beare, 2008) as well as on the ET (Sorbjan, 1996, 1997; Beare et al., 2006; Pino et al., 2006). More recently, diurnal cycle studies were conducted by Kumar et al. (2006) and Basu et al. (2008).' instead of lines 46-55.

[2.6] We decided to incorporate WT modelling for an independent publication. Therefore, we performed an additional idealised ABL simulation over homogeneous surface with a geostrophic wind of $u = 10 \text{ m s}^{-1}$ and $v = 0 \text{ m s}^{-1}$. This precursor simulation is used as background, initial and inflow condition of $u$, $v$, $w$, and $\Theta$ in the WT simulations for different regimes (CBL, ET, SBL, MT). 'To our knowledge, this is the first study which investigates the influence of a full diurnal cycle, including the ET and the MT, on the wake of a single WT.' This represents the unique contribution of the manuscript.

Further, we expand the idealised ABL simulation to 54 h to also include the second diurnal cycle, investigating the difference of the WT wakes between the diurnal cycle of day 1 and day 2.

Referee Further, the correspondence of these simulations to the BLLAST observations is questionable. The authors compare their potential temperature profiles to the observed potential temperature profiles at only three points in the diurnal cycle (one profile is used for initialization, three for evaluation). No data other than soundings is used for evaluation although BLLAST included considerable instrumental deployments.[3.1] They use two tuning parameters (subsidence and radiative cooling) to achieve approximate agreement with the profiles (but, as noted below, the authors do not refer to previous work on LES with subsidence). How should a reader develop confidence in the selection of subsidence rate and cooling rate? Are there any observations that support these choices of subsidence or radiative cooling?[3.2] Second, the authors modify the winds in their simulations substantially from the observed 3 m s-1 to 10 m s-1 (a factor of three!) but still suggest that their simulations compare well to the BLLAST observations. It would be a cleaner comparison to first match both the winds and the thermodynamics (so that they can validate simulations with observed fluxes, aircraft data, etc.). Later, once the reader trusts the simulations, the authors could increase the winds if necessary. As the simulations stand right now, they are not really based on any observations with so many tuning parameters and vastly different winds.[3.3] If the authors really require winds of 10 m s-1, they should find

another experiment (CASES-99? Numerous studies from Cabauw?) that can provide adequate data for validation.[3.4]

Author [3.1] We tried to compare the temporal and spatial evolution of $u$ and $v$ with sodar measurement for $z \leq 800$ m and with UHF wind profiler data for $z > 800$ m. This comparison does not lead to a satisfying agreement after a few hours of simulation. In our opinion, it is very difficult to fit the observed values of the wind speed and the wind direction with the LES. This is for two reasons. First, our numerical simulation strategy. We initialise with the 0000 UTC radiosonde data, however, we do not nudge during the simulation to measured wind profiles. Second, mesoscale processes. The dynamics of the atmosphere might be dominated by mesoscales processes like mountain-valley flows, which are not considered in the LES. We considered this in the new manuscript as: 'Considering 23 h of simulation with the limited prescribed external forcings, the colder temperature in the lowest levels can be caused by additional large-scale effects, which are not included in our simulation, e.g. colder air close to the surface advecting from the mountains as part of the mountain-plain circulation. '

In addition, we tried to compare the temporal and spatial evolution of $\theta$ with radiometer measurements. Unfortunately, from the 01.07.11 to the 05.07.11 there are not data available.

In summary, we arrive at the conclusion that the validation with the BLLAST data of $\theta$ from the radiosonde launch gives confidence in the ability of the geophysical flow solver EULAG to simulate the diurnal cycle of idealised ABL simulations, which is an important step towards the investigation of the diurnal variation of different atmospheric variables relevant in wind energy research ($u$, $I$) and the investigation of the wake structure in WT simulations. We stated this in the new version of the manuscript as: Introduction: 'We simulate the complete diurnal cycle of one day by using observations from the BLLAST field campaign and the LES model EULAG. The chosen day was characterized by a surface driven weather situation with minimum larger scale disturbances. Therefore, it is well-suited to validate the geophysical flow solver EULAG.' Conclusion: 'The validation of a full diurnal cycle of the BLLAST field experiment with our LES model EULAG gives confidence in ABL simulations. ' For this purpose, we decide that we do not need a more detail investigation for the main statement of this manuscript, the impact of the diurnal cycle of the atmospheric boundary layer on wind turbine wakes.

[3.2] We included references of previous work on LES with subsidence of Mirocha and Kosović (2010) and Bellon and Stevens (2012) to Blay-Carreras et al. (2014) (line 212) and Mazzitelli et al. (2014) (line 194) in the following sentence: 'Subsidence dumps vertical motions at scales larger than the mixed layer height, especially during convective conditions, limiting the growth of the ABL and influencing temperature and turbulence in simulations of the ABL (Mirocha and Kosović, 2010; Blay-Carreras et al., 2014; Mazzitelli et al., 2014).'

Further, the values of subsidence and radiative cooling are calculated with the help of the radiosonde profiles. It is explained now in more detail in the text: 'The subsidence velocity is calculated from the ABL height of the potential temperature profiles from the four radiosonde ascents at 0000 UTC, 1100 UTC, 1658 UTC and 2254 UTC to a value of $w_{sub} = -10 \, \mathrm{mm \, s^{-1}}$. A radiative cooling of the atmosphere of $-2 \, \mathrm{K \, d^{-1}}$ arises from the difference of the simulated potential temperature profile in a different simulation with exactly the same set-up as the BLLAST ABL simulation, however, without radiative cooling. The values of the subsidence velocity and the radiative cooling are comparable to values of Bellon and Stevens (2012).' instead of lines 273-280.

[3.3] We rerun the BLLAST ABL simulation initialised with the measured wind profile from the 0000 UTC radiosonde launch for u and v in addition to $\Theta$ instead of a geostrophic wind of $u = 10 \, \mathrm{m \, s^{-1}}$ and $v = 0 \, \mathrm{m \, s^{-1}}$ and updated the results in chapter 3. In the new version of chapter 3 we investigate the time and space variation of the potential temperature, the individual TKE budget terms integrated over the height of the ABL and as vertical evolution and the temporal evolution of the potential temperature (old Fig. 1). (See Fig. 1 below.)

[3.4] We decided to performed an additional idealised ABL simulation with a geostrophic wind of $u = 10 \, \mathrm{m \, s^{-1}}$ and $v = 0 \, \mathrm{m \, s^{-1}}$ for the investigation of atmospheric variables which are relevant in wind energy research ($u$, $I$).

Referee  Finally, the authors do carry out a small ensemble of simulations with varying canopies and obstacles in the flow, emphasizing the novelty of surface heterogeneity in LES of the ABL. Again, they have not reviewed previous work in this area (Belcher et al. 2003; Kang et al. 2012, among others), nor have they supplied sufficient details on their implementation of the immersed boundary method for readers to understand if this is a novel contribution. Nevertheless, there could be novel aspects to this part of their study that would justify a publication on their LES.[4.1]

Author  [4.1] We skip the simulations with heterogeneous surface in this manuscript (section 5), because a detailed investigation of the ABL simulation over heterogeneous surface together with the corresponding WT simulations over heterogeneous surface would result in a rather long manuscript. Future research will include this part. As novel aspect we included the investigation of the influence of the ET and the MT on WT wakes and LESs investigating the wake of a single WT for the CBL and the SBL resulting from the same diurnal cycle.

**Specific Comments**

Referee: 27  "its minimum level" should be "their minimum level" (referring to surface fluxes)

Author:  We changed it.

Referee: 35  should read "The CBL is not only influenced from below, but also from above, as entrainment processes incorporate..."

Author:  We skip this particular phrase. Instead we included lines 22-44 in section 3 as part of the evaluation of the BLLAST ABL simulation and modified it according to: 'For the evaluation of the BLLAST ABL simulation, the sensible heat flux, the potential temperature, and the TKE budget are shown in Fig. 1. The structure of the potential temperature and the TKE budget are strongly influenced by the diurnal cycle enforced by the prescribed sensible heat flux. For rather low surface flux values in $t \in [0\ h, 5\ h]$, the ABL in Fig. 1($b$) consists of an SBL capped by a neutrally stratified layer, the RL. An increase of the surface fluxes at t = 5 h from their minimum level initiates the onset of the MT, which is related to a warming of the surface in Fig. 1($b$). This generates thermals and the turbulent eddies increase in size and strength and start to form a fully convective layer, which continues to grow throughout the morning eroding the stable layer from below and incorporating the RL in a process called free encroachment (Sorbjan, 2004) (Fig. 1($b$)). This results in a fully developed CBL, characterised by the domination of buoyancy over shear in Figs. 1($c$) and 2($a$) (Stull, 1988; Beare, 2008). The decrease of the surface fluxes approaching their minimum level represents the ET. During the ET, the decaying CBL merges into the SBL, which is shear driven, because the eddies have less energy in comparison to the CBL due to negative surface flux values (Stull, 1988; Beare, 2008). The domination of shear over buoyancy is not visible in Fig. 1($c$) because the individual TKE budget terms are approximately an order of magnitude smaller at night in comparison to the daytime situation. A comparison of the vertical evolution of the horizontal average of the TKE budget in Fig. 2($b$), however, confirms this statement. Further, the height of the ABL during the diurnal cycle simulation corresponds to the stronger stratification in Fig. 1($b$), which is limited by the imposed large-scale subsidence.'

Referee: 37  updraft and downdraft instead of "updraught" etc.

Author:  We skip this part as it is not relevant in the main part of the manuscript.

Referee: 40  why are these citations not in chronological order?

Author:  This happened unintentionally. We skip this part as it is not relevant in the main part of the manuscript.

Referee: 46-55  these lists of citations are not put into context for the reader

Author:  We only aim to give an overview of ABL simulations by listing a few of the numerous papers on the varies aspects:

- SBL: Nieuwstadt (1984), Carlson and Stull (1986), Mahrt (1998)
- RL: Balsley et al. (2008), Wehner et al. (2010)

- CBL:

  - coherent structures: Schmidt and Schumann (1989)

  - entrainment: Sorbjan (1996), Sullivan et al. (1998), Conzemius and Fedorovich (2007)

  - shear: Moeng and Sullivan (1994), Fedorovich et al. (2001), Pino et al. (2003)

- MT: Sorbjan (2007), Beare (2008)

- ET: Sorbjan (1996), Sorbjan (1997), Beare et al. (2006), Pino et al. (2006)

A connection to our study follows in the next paragraph of the new version of the manuscript: 'We simulate the complete diurnal cycle of one day by using observations from the BLLAST field campaign and the LES model EULAG. The chosen day was characterized by a surface driven weather situation with minimum larger scale disturbances. Therefore, it is well-suited to validate the geophysical flow solver EULAG. Further, we perform a simulation of an idealised ABL throughout two full diurnal cycles with periodic horizontal boundary conditions. This is for two reasons. First, to investigate the diurnal variation of different atmospheric variables relevant in wind energy research. Second, as real-time atmospheric inflow condition in simulations of a single WT with open horizontal boundary conditions for the investigation of the impact of the CBL, the ET, the SBL and the MT in the course of a day on the wake structure. To our knowledge, this is the first study which investigates the influence of a full diurnal cycle, including the ET and the MT, on the wake of a single WT. '

Referee: 58 this research question is not particularly innovative as LES of the diurnal cycle has been carried out (Kumar et al. 2006; Basu et al. 2008) (and these papers should be cited)

Author: Our LES of the diurnal cycle performed with the BLLAST data from the radiosonde launch is not innovative, it only gives confidence in the ability of the geophysical flow solver EULAG to simulate the diurnal cycle of idealised ABL simulations. This is an important step to perform an idealised ABL simulation with a geostrophic wind of $u = 10$ m s$^{-1}$ and $v = 0$ m s$^{-1}$ for the investigation of atmospheric variables which are relevant in wind energy research ($u$, $I$). A reliable idealised ABL simulation is also important for the investigation of the wake structure in WT simulations. This is stated in the new version of the manuscript: 'We simulate the complete diurnal cycle of one day by using observations from the BLLAST field campaign and the LES model EULAG. The chosen day was characterized by a surface driven weather situation with minimum larger scale disturbances. Therefore, it is well-suited to validate the geophysical flow solver EULAG. Further, we perform a simulation of an idealised ABL throughout two full diurnal cycles with periodic horizontal boundary conditions. This is for two reasons. First, to investigate the diurnal variation of different atmospheric variables relevant in wind energy research. Second, as real-time atmospheric inflow condition in simulations of a single

WT with open horizontal boundary conditions for the investigation of the impact of the CBL, the ET, the SBL and the MT in the course of a day on the wake structure.'

Further, we changed the focus of the manuscript to WT simulations for CBL, SBL, ET and MT regimes. The investigation of the influence of a full diurnal cycle, including the ET and the MT, on the wake of a single WT is the innovative part in the new version of the manuscript. 'To our knowledge, this is the first study which investigates the influence of a full diurnal cycle, including the ET and the MT, on the wake of a single WT.' (Also in [2.5], [2.6] and [3.4].)

Referee: 61   Why is a recent paper on wind energy research (Emanuel et al. 2015) referred to for discussion of the diurnal cycle of the ABL when many other older and more detailed studies of the diurnal cycle of the ABL for wind energy applications could be considered? should cite (Barthelmie et al. 1996; Walter et al. 2009; Storm and Basu 2010; Rhodes and Lundquist 2013, among others)

Author:   We changed the introduction. This part is no longer in the manuscript in this form. Further, we included Walter et al. (2009) and Rhodes and Lundquist (2013) in section 3 of the new manuscript in the following paragraph: 'A quantitative comparison of the horizontally averaged $\Theta$ profiles with the radiosonde measurements from the BLLAST campaign is shown in Fig. 3 (new version). The simulation is initialised with a fit to the observed 0000 UTC potential temperature profile. The potential temperature structure evolves towards a convective profile in the first eleven hours (Stull, 1988; Kumar et al., 2006; Basu et al., 2008; Beare, 2008; Walter et al., 2009; Rhodes and Lundquist, 2013). The decrease of the ABL height is induced by the imposed large-scale subsidence. After 17 h, the simulated $\Theta$ profile fits well with the observation for $z < 1300$ m. Above the inversion, modelled and observed profiles differ, as other mesoscale influences are not taken into account in the LES. After 23 h, a stable stratification has established again (Stull, 1988; Kumar et al., 2006; Basu et al., 2008; Beare, 2008; Walter et al., 2009; Rhodes and Lundquist, 2013). Considering 23 h of simulation with the limited prescribed external forcings, the colder temperature in the lowest levels can be caused by additional large-scale effects, which are not include in our simulation, e.g. colder air close to the surface advecting from the mountains as part of the mountain-plain circulation.'

Referee: 63-64   several papers have investigated the impact of atmospheric stability on wind turbine wakes (Aitken et al. 2014; Bhaganagar and Debnath 2015; Mirocha et al. 2015, for example)

Author:   The papers of Aitken et al. (2014), Bhaganagar and Debnath (2015) and Mirocha et al. (2014) are included in the WT part of the introduction and also in the WT chapter (chapter 5 of new version). We further include: Baker and Walker (1984); Magnusson and Smedman (1994); Barthelmie and Jensen (2010); Porté-Agel et al. (2010); Naughton et al. (2011); Wu and Porté-Agel (2011, 2012); Zhang et al. (2013); Iungo and Porté-Agel (2014); Hancock and Pascheke

(2014); Bhaganagar and Debnath (2014); Gomes et al. (2014); Abkar and Porté-Agel (2014); Englberger and Dörnbrack (2016); Abkar et al. (2016)

Referee: 73  wind turbine loading should refer to (Sathe et al. 2013)

Author:  We included it: 'Atmospheric turbulence has an impact on the power produced by a WT and on the turbine loading (Sathe et al., 2013) because it affects the streamwise extension of the wake, the magnitude of the velocity deficit and the turbulence in the wake.'

Referee: 79  the comment that "most papers assume NBL" is not consistent with the subsequent citations focusing on the SBL and CBL

Author:  Some recent LES studies start to investigate the impact of different atmospheric stratifications on the wake flow. Up to now, there are a numerous of studies. However, there are many more numerical studies assuming an neutral boundary layer (NBL) as atmospheric state. We changed it from 'most papers assume NBL' to 'most of the numerical simulations of a WT assume an NBL' to make our point more clear, trying to avoid any misunderstanding. ('However, most of the numerical simulations of a WT assumed a NBL (Porté-Agel et al., 2010; Wu and Porté-Agel, 2011; Naughton et al., 2011), even though the entrainment of energy and momentum into the wake region and the resulting wake structure strongly depend on the level of atmospheric turbulence in the upstream region of a WT.')

Referee: 87  The discussion states that no other papers have explored how the complete diurnal cycle influences wind turbine wakes, but that is not the point of this present manuscript. This paper focuses on parameters relevant to wind energy applications, which is different from wind turbine wakes. It feels like the authors are trying to define a specific niche, not very accurately, and without correspondence to the goal of the present paper. The authors should think carefully about the specific novel contributions of the specific work presented here, and emphasize those contributions, not necessarily the larger project.

Author:  Following your comment, we decided to incorporate WT modelling for an independent publication and novel contribution with the investigation of the influence of a full diurnal cycle, including the ET and the MT, on the wake of a single WT. These two points are the novel contributions in the new version of the manuscript.

Referee: 144  the subgridscale model is very important but is not discussed in any detail here. Is it a TKE-based model? More details are necessary.

Author:  See $^{2.4}$.

Referee: 167  "satisfactory" rather than "satisfying"

Author:  We changed it.

Referee: 175 Please explain that "MT" stands for "morning transition" and similarly for "ET"

Author: It was included in lines 19-20. We changed it, try to make it more clear with: 'These transition periods are referred to as morning transition (MT) and evening transition (ET) and are defined following Grimsdell and Angevine (2002) as the time period in which the sensible heat flux changes sign.'

Referee: 194 "descent" rather than "decent"; check grammar in this sentence,

Author: This was a typo. We eliminate it and changed the sentence to :'For a realistic simulation of the diurnal cycle (ABL height, temperature and turbulence evolution), it is important to model and integrate additional external forcings representing mesoscale and synoptic scale processes into the LES. We select subsidence and radiative cooling of the atmosphere as dominant large-scale external forcing processes.'

Referee: 194 discussion of subsidence is interesting but very focused on recent work. Please also refer to previous work on subsidence such as (Mirocha and Kosović 2009). In fact, given that subtle changes in the specification of subsidence can significantly change surface fluxes and boundary-layer height, how can the authors justify their choice of $w_{sub}$ (and the resulting choice for radiative cooling?)

Author: It is explained in more detail in the new version of the manuscript. (see detailed answer in [3.2])

Referee: 233 more explanation or references are required for the "immersed boundary method" – many choices for how to implement the immersed boundary method are possible.

Author: We eliminate the immersed boundary method. (see detailed answers in [2.3] and [4.1])

Referee: 258 validation of LES is carried out only against radiosondes? What about flux measurements and the multiple other measurements possible during BLLAST?

Author: We calculate the sensible heat flux with the flux measurements from the tower, as stated in: 'The flux measurements are taken from a 60 m tower with values at 30 m, 45 m and 60 m altitude. The sensible heat flux (SHF) at the surface is calculated via an e-folding scale of 300 m and the measurements at the heights $z$ according to $\mathrm{SHF}_{z=0} = \mathrm{SHF}_z \exp(-z / 300 \text{ m})$.' (old version: lines 268-271). For the other possible measurements see detailed answer in [3.1].

Referee: Figure 1 difficult to interpret for red-green color-blind readers. Can the lines be thicker and labelled directly instead of relying on the legend? Are axis labels consistent with ACP requirements?

Author: We try to improve it: We made the lines thicker. We also make the axis labels consistent with the ACP requirements. See Fig. 1 below.

Referee: 271 should be "W m-2" rather than "W m2"

Author: This was a typo. We corrected it.

Referee: 283 it appears that the authors are attempting to match the thermodynamic aspects of the observations but for a difference range of wind speeds (10 m/s instead of the observed 3 m/s). Wouldn't the thermodynamic profiles have changed with different winds? The utility of the comparison to observations seems lacking, and it is not clear that "validation" describes what is happening here.

Author: We rerun the simulation with the measures wind speeds (see detailed answer in [3.3]). Comparing Fig. 1 with the corresponding new figure from below (Fig. 3 in new version of the manuscript), the thermodynamic profiles are rather comparable for wind speeds of 10 m s$^{-1}$ and 3 m s$^{-1}$. For a more detailed comparison to other observations see detailed answer in [3.1].

Instead of 'validation' we use 'evaluation' and 'qualitative and comparison' in the new version of the manuscript in: 'For the evaluation of the BLLAST ABL simulation, ...', 'A quantitative comparison of the horizontally averaged $\Theta$ profiles with the radiosonde measurements from the BLLAST campaign ...' and 'The qualitative comparison of the temporal evolution of the potential temperature and the analysis of the TKE budget together with the quantitative comparison of the LES results with the potential temperature profiles of radiosonde measurements at 1100 UTC, 1700 UTC and 2300 UTC ...'

Referee: 300-315 this "validation" is really a qualitative comparison. Can quantitative metrics be calculated?

Author: We perform a quantitative comparison of the simulated $\Theta$ profiles with radiosonde measurements, as stated in the new version of the manuscript: 'A quantitative comparison of the horizontally averaged $\Theta$ profiles with the radiosonde measurements from the BLLAST campaign is shown in Fig. 1. The simulation is initialised with a fit to the observed 0000 UTC potential temperature profile. The potential temperature structure evolves towards a convective profile in the first eleven hours (Stull, 1988; Kumar et al., 2006; Basu et al., 2008; Beare, 2008; Walter et al., 2009; Rhodes and Lundquist, 2013). The decrease of the ABL height is induced by the imposed large-scale subsidence. After 17 h, the simulated $\Theta$ profile fits well with the observation for z < 1300 m. Above the inversion, modelled and observed profiles differ, as other mesoscale influences are not taken into account in the LES. After 23 h, a stable stratification has established again (Stull, 1988; Kumar et al., 2006; Basu et al., 2008; Beare, 2008; Walter et al., 2009; Rhodes and Lundquist, 2013). Considering 23 h of simulation with the limited prescribed external forcings, the colder temperature in the lowest levels can be caused by additional large-scale effects, which are not included in our simulation, e.g. colder air close to the surface advecting from the mountains as part of the mountain-plain circulation.' instead of lines 295-315

For additional quantitative metrics a much more detailed investigation of the BLLAST ABL would be necessary. This, however, goes beyond the scope of this manuscript, as it is not of major importance for the main part of this manuscript (diurnal cycle impact of idealised ABL on variables $u$ and $I$ and diurnal cycle impact on WT wakes).

Referee: 311-314   speculation on what tweaks could improve model performance are not appropriate: either make the changes and evaluate the simulations or avoid commenting on possibilities.

Author:   We eliminate it.

Referee: 316-319   sentence unclear

Author:   We rewrote it to: 'The qualitative comparison of the temporal evolution of the potential temperature and the analysis of the TKE budget together with the quantitative comparison of the LES results with the potential temperature profiles of radiosonde measurements at 1100 UTC, 1700 UTC and 2300 UTC give confidence in the ability of the geophysical flow solver EULAG to simulate the diurnal cycle of the idealised ABL simulation in the following.'

Referee: 334-349   these observations are not particularly novel or innovative for LES of a CBL.

Author:   We skip them. Instead we included a detailed investigation of $u$ and $I$ in chapter 4 of the new version of the manuscript.

Referee: 356   Please refer to wind energy industry references for assertions of "typical" parameters. The Global Wind Energy Council has appropriate statistics.

Author:   We avoid the word 'typical' in the new version of the manuscript. We define a WT with a diameter of 100 m and a hub height of 100 m.

Referee: 366   The variability of the wind profile between daytime and nighttime is well known and documented in the literature already (Barthelmie et al. 1996; Walter et al. 2009; Rhodes and Lundquist 2013)

Author:   We added a few of the relevant papers. A detailed description in 61 above.

Referee: 370-417   The discussion of the TKE budget does not introduce novel insights compared to previous studies such as (Beare et al. 2006)

Author:   We need the discussion of the TKE budget for the explanation of the WT wakes. No novel insights are intended in this part. Our results only confirm that the simulated TKE budget terms are representative for a typical ABL evolution.

Referee: 440   The main conclusion seems to be the effect of obstacles on shear. Again, the novelty of this result is questionable (see previous work, such as (Belcher et al. 2003))

Author:   See detailed answer in [4.1].

**References**

Abkar, M. and Porté-Agel, F.: The effect of atmospheric stability on wind-turbine wakes: A large-eddy simulation study, in: Journal of Physics: Conference Series, vol. 524, p. 012138, IOP Publishing, doi:10.1088/1742-6596/524/1/012138, 2014.

Abkar, M., Sharifi, A., and Porté-Agel, F.: Wake flow in a wind farm during a diurnal cycle, Journal of Turbulence, 17, 420–441, doi:10.1080/14685248.2015.1127379, 2016.

Aitken, M. L., Kosović, B., Mirocha, J. D., and Lundquist, J. K.: Large eddy simulation of wind turbine wake dynamics in the stable boundary layer using the Weather Research and Forecasting Model, Journal of Renewable and Sustainable Energy, 6, 033 137, doi:10.1063/1.4885111, 2014.

Baker, R. W. and Walker, S. N.: Wake measurements behind a large horizontal axis wind turbine generator, Solar Energy, 33, 5–12, doi:10.1016/0038-092X(84)90110-5, 1984.

Balsley, B. B., Svensson, G., and Tjernström, M.: On the scale-dependence of the gradient Richardson number in the residual layer, Bound-Lay Meteorol, 127, 57–72, doi:10.1007/s10546-007-9251-0, 2008.

Barthelmie, R. J. and Jensen, L.: Evaluation of wind farm efficiency and wind turbine wakes at the Nysted offshore wind farm, Wind Energy, 13, 573–586, doi:10.1002/we.408, 2010.

Basu, S., Vinuesa, J.-F., and Swift, A.: Dynamic LES modeling of a diurnal cycle, Journal of Applied Meteorology and Climatology, 47, 1156–1174, doi:10.1175/2007JAMC1677.1, 2008.

Beare, R. J.: The Role of Shear in the Morning Transition Boundary Layer, Bound-Lay Meteorol, 129, 395–410, doi:10.1007/s10546-008-9324-8, 2008.

Beare, R. J., Macvean, M. K., Holtslag, A. A. M., Cuxart, J., Esau, I., Golaz, J.-C., Jimenez, M. A., Khairoutdinov, M., Kosović, B., Lewellen, D., Lund, T. S., Lundquist, J. K., Mccabe, A., Moene, A. F., Noh, Y., Raasch, S., and Sullivan, P.: An Intercomparison of Large-Eddy Simulations of the Stable Boundary Layer, Bound-Lay Meteorol, 118, 247–272, doi:10.1007/s10546-004-2820-6, 2006.

Bellon, G. and Stevens, B.: Using the sensitivity of large-eddy simulations to evaluate atmospheric boundary layer models, Journal of the Atmospheric Sciences, 69, 1582–1601, doi:10.1175/JAS-D-11-0160.1, 2012.

Bhaganagar, K. and Debnath, M.: Implications of Stably Stratified Atmospheric Boundary Layer Turbulence on the Near-Wake Structure of Wind Turbines, Energies, 7, 5740–5763, doi:10.3390/en7095740, 2014.

Bhaganagar, K. and Debnath, M.: The effects of mean atmospheric forcings of the stable atmospheric boundary layer on wind turbine wake, Journal of Renewable and Sustainable Energy, 7, 013 124, doi:10.1063/1.4907687, 2015.

Blay-Carreras, E., Pino, D., Vilà-Guerau de Arellano, J., van de Boer, A., De Coster, O., Darbieu, C., Hartogensis, O., Lohou, F., Lothon, M., and Pietersen, H.: Role of the residual layer and large-scale subsidence on the development and evolution of the convective boundary layer, Atmos Chem Phys, 14, 4515–4530, doi:10.5194/acp-14-4515-2014, 2014.

Carlson, M. A. and Stull, R. B.: Subsidence in the nocturnal boundary layer, J Clim Appl Meteorol, 25, 1088–1099, doi:10.1175/1520-0450(1986)025<1088:SITNBL>2.0.CO;2, 1986.

Conzemius, R. and Fedorovich, E.: Bulk models of the sheared convective boundary layer: Evaluation through large eddy simulations, J Atmos Sci, 64, 786–807, doi:10.1175/JAS3870.1, 2007.

Deardorff, J. W.: Three-dimensional numerical study of the height and mean structure of a heated planetary boundary layer, Bound-Lay Meteorol, 7, 81–106, doi:10.1007/BF00224974, 1974a.

Deardorff, J. W.: Three-dimensional numerical study of turbulence in an entraining mixed layer, Bound-Lay Meteorol, 7, 199–226, doi:10.1007/BF00227913, 1974b.

Englberger, A. and Dörnbrack, A.: Impact of atmospheric boundary-layer turbulence on wind-turbine wakes: A numerical modelling study, Bound-Lay Meteorol, in review, 2016.

Fedorovich, E., Nieuwstadt, F., and Kaiser, R.: Numerical and laboratory study of a horizontally evolving convective boundary layer. Part I: Transition regimes and development of the mixed layer, J Atmos Sci, 58, 70–86, doi:10.1175/1520-0469(2001)058<0070:NALSOA>2.0.CO;2, 2001.

Gomes, V. M. M. G. C., Palma, J. M. L. M., and Lopes, A. S.: Improving actuator disk wake model, in: The science of making torque from wind. Conference series, vol. 524, p. 012170., doi:10.1088/1742-6596/524/1/012170, 2014.

Grimsdell, A. W. and Angevine, W. M.: Observations of the afternoon transition of the convective boundary layer, J Appl Meteorol, 41, 3–11, doi:10.1175/1520-0450(2002)041<3C0003:OOTATO>3E2.0.CO;2, 2002.

Hancock, P. E. and Pascheke, F.: Wind-tunnel simulation of the wake of a large wind turbine in a stable boundary layer: Part 2, the wake flow, Boundary-layer meteorology, 151, 23–37, doi:10.1007/s10546-013-9887-x, 2014.

Iungo, G. V. and Porté-Agel, F.: Volumetric lidar scanning of wind turbine wakes under convective and neutral atmospheric stability regimes, Journal of Atmospheric and Oceanic Technology, 31, 2035–2048, doi:10.1175/JTECH-D-13-00252.1, 2014.

Kumar, V., Kleissl, J., Meneveau, C., and Parlange, M. B.: Large-eddy simulation of a diurnal cycle of the atmospheric boundary layer: Atmospheric stability and scaling issues, Water resources research, 42, doi:10.1029/2005WR004651, 2006.

Magnusson, M. and Smedman, A.: Influence of atmospheric stability on wind turbine wakes, Wind Engineering, 18, 139–152, doi:10.1063/1.4913695, 1994.

Mahrt, L.: Nocturnal boundary-layer regimes, Bound-Lay Meteorol, 88, 255–278, doi:10.1023/A:1001171313493, 1998.

Mazzitelli, I. M., Cassol, M., Miglietta, M. M., Rizza, U., Sempreviva, a. M., and Lanotte, a. S.: The role of subsidence in a weakly unstable marine boundary layer: a case study, Nonlinear Proc Geoph, 21, 489–501, doi:10.5194/npg-21-489-2014, 2014.

Mirocha, J. D. and Kosović, B.: A large-eddy simulation study of the influence of subsidence on the stably stratified atmospheric boundary layer, Boundary-layer meteorology, 134, 1–21, doi:10.1007/s10546-009-9449-4, 2010.

Mirocha, J. D., Kosović, B., Aitken, M. L., and Lundquist, J. K.: Implementation of a generalized actuator disk wind turbine model into the weather research and forecasting model for large-eddy simulation applications, Journal of Renewable and Sustainable Energy, 6, 013 104, doi:10.1063/1.4861061, 2014.

Moeng, C.-H. and Sullivan, P. P.: A comparison of shear-and buoyancy-driven planetary boundary layer flows, J Atmos Sci, 51, 999–1022, doi:10.1175/1520-0469(1994)051<0999:ACOSAB>2.0.CO;2, 1994.

Naughton, J. W., Heinz, S., Balas, M., Kelly, R., Gopalan, H., Lindberg, W., Gundling, C., Rai, R., Sitaraman, J., and Singh, M.: Turbulence and the isolated wind turbine, in: 6th AIAA Theoretical Fluid Mechanics Conference, pp. 1–19., Honolulu, Hawaii, doi:10.2514/6.2011-3612, 2011.

Nieuwstadt, F. T.: The turbulent structure of the stable, nocturnal boundary layer, J Atmos Sci, 41, 2202–2216, doi:10.1175/1520-0469(1984)041<2202:TTSOTS>2.0.CO;2, 1984.

Pino, D., Vilà-Guerau de Arellano, J., and Duynkerke, P. G.: The contribution of shear to the evolution of a convective boundary layer, J Atmos Sci, 60, 1913–1926, doi:10.1175/1520-0469(2003)060<1913:TCOSTT>2.0.CO;2, 2003.

Pino, D., Jonker, H. J., De Arellano, J. V.-G., and Dosio, A.: Role of shear and the inversion strength during sunset turbulence over land: characteristic length scales, Bound-Lay Meteorol, 121, 537–556, doi:10.1007/s10546-006-9080-6, 2006.

Porté-Agel, F., Lu, H., and Wu, Y.-t.: A large-eddy simulation framework for wind energy applications, in: The Fifth International Symposium on Computational Wind Engineering, 2010.

Prusa, J. M., Smolarkiewicz, P. K., and Wyszogrodzki, A. A.: EULAG, a computational model for multiscale flows, Comput Fluids, 37, 1193–1207, doi:10.1016/j.compfluid.2007.12.001, 2008.

Rhodes, M. E. and Lundquist, J. K.: The effect of wind-turbine wakes on summertime US Midwest atmospheric wind profiles as observed with ground-based doppler lidar, Boundary-Layer Meteorology, 149, 85–103, doi:10.1007/s10546-013-9834-x, 2013.

Sathe, A., Mann, J., Barlas, T., Bierbooms, W., and Bussel, G.: Influence of atmospheric stability on wind turbine loads, Wind Energy, 16, 1013–1032, doi:10.1002/we.1528, 2013.

Schmidt, H. and Schumann, U.: Coherent structure of the convective boundary layer derived from large-eddy simulations, Journal of Fluid Mechanics, 200, 511–562, doi:10.1017/S0022112089000753, 1989.

Smolarkiewicz, P. K. and Margolin, L. G.: MPDATA: A Finite-Difference Solver for Geophysical Flows, J Comput Phys, 140, 459–480, doi:10.1006/jcph.1998.5901, 1998.

Sorbjan, Z.: Effects caused by varying the strength of the capping inversion based on a large eddy simulation model of the shear-free convective boundary layer, J Atmos Sci, 53, 2015–2024, doi:10.1175/1520-0469(1996)053<2015:ECBVTS>2.0.CO;2, 1996.

Sorbjan, Z.: Decay of convective turbulence revisited, Bound-Lay Meteorol, 82, 503–517, doi:10.1023/A:1000231524314, 1997.

Sorbjan, Z.: Large-eddy simulations of the baroclinic mixed layer, Bound-Lay Meteorol, 112, 57–80, doi:10.1023/B:BOUN.0000020161.99887.b3, 2004.

Sorbjan, Z.: A numerical study of daily transitions in the convective boundary layer, Bound-Lay Meteorol, 123, 365–383, doi:10.1007/s10546-006-9147-4, 2007.

Stull, R. B.: An Introduction of Boundary Layer Meteorology, Dordrecht, Kluwer Academic, 1988.

Sullivan, P. P., Moeng, C.-H., Stevens, B., Lenschow, D. H., and Mayor, S. D.: Structure of the Entrainment Zone Capping the Convective Atmospheric Boundary Layer, J Atmos Sci, 55, 3042–3064, doi:10.1175/1520-0469(1998)055<3042:SOTEZC>2.0.CO;2, 1998.

Walter, K., Weiss, C. C., Swift, A. H., Chapman, J., and Kelley, N. D.: Speed and direction shear in the stable nocturnal boundary layer, Journal of Solar Energy Engineering, 131, 011 013, doi:10.1115/1.3035818, 2009.

Wehner, B., Siebert, H., Ansmann, A., Ditas, F., Seifert, P., Stratmann, F., Wiedensohler, A., Apituley, A., Shaw, R., Manninen, H., et al.: Observations of turbulence-induced new particle formation in the residual layer, Atmos Chem Phys, 10, 4319–4330, doi:10.5194/acp-10-4319-2010, 2010.

Wu, Y. T. and Porté-Agel, F.: Large-Eddy Simulation of Wind-Turbine Wakes: Evaluation of Turbine Parametrisations, Bound-Lay Meteorol, 138, 345–366, doi:10.1007/s10546-010-9569-x, 2011.

Wu, Y. T. and Porté-Agel, F.: Atmospheric Turbulence Effects on Wind-Turbine Wakes: An LES Study, Energies, 5, 5340–5362, doi:10.3390/en5125340, 2012.

Zhang, W., Markfort, C. D., and Porté-Agel, F.: Wind-Turbine Wakes in a Convective Boundary Layer: A Wind-Tunnel Study, Bound-Lay Meteorol, 146, 161–179, doi:10.1007/s10546-012-9751-4, 2013.

[Figure]

**Figure 1.** The temporal evolution of the horizontal average of $\Theta(z)$ (solid lines) and the corresponding radiosonde measurements (dashed lines) at 0 h, 11 h, 17 h and 23 h for the lowest 2 km.

---

## Author Comment (AC2) · 28 Aug 2016

Manuscript prepared for J. Name
with version 2015/09/17 7.94 Copernicus papers of the LaTeX class copernicus.cls.
Date: 28 August 2016

**Comments of Anonymous Referee 2 on 'The impact of the diurnal cycle of the atmospheric boundary layer on physical variables relevant for wind energy applications' by A. Englberger and A. Dörnbrack**

Antonia Englberger and Andreas Dörnbrack

Antonia.Englberger@dlr.de

First of all, we would like to acknowledge the comments. They were very helpful for improving the manuscript. Below the general comments and the specific comments made by referee 2 are discussed in detail.

Additionally, we changed the title from 'The impact of the diurnal cycle of the atmospheric boundary layer on physical variables relevant for wind energy applications' to 'Impact of the diurnal cycle of the atmospheric boundary layer on wind turbine wakes: A numerical modelling study'.

**General comments**

Referee  The paper discusses an LES study of a full diurnal cycle of the ABL that is validated with data from the BLLAST Field program. The study is an intermediate step toward a full LES model that incorporates wind turbine modeling. I have three general concerns regarding the manuscript. The first concern is that the paper does not appear to be sufficiently novel to stand as an independent publication. The contributions of the paper, in their current state, seem more appropriate as a chapter of a larger study.[1.1] This is due to much of the introduction discussing turbine modeling, something that is not actually modeled in the paper. The introduction should put less emphasis on turbine modeling and more emphasis on literature relevant to modeling the full diurnal cycle with realistic boundary conditions. A quick Google search reveals that a number of previous LES publications have studied the full diurnal cycle. What is unique and novel about your approach?[1.2]

Author  [1.1] Considering your comment, we decided to incorporate wind turbine (WT) modelling for an independent publication. Therefore, we performed an additional idealised atmospheric boundary layer (ABL) simulation over homogeneous surface with a geostrophic wind of $u = 10$ m s$^{-1}$ and $v = 0$ m s$^{-1}$. This precursor simulation is used as background, initial and inflow condition of $u$, $v$, $w$, and $\Theta$ in the WT simulations for different regimes (convective boundary layer (CBL), evening transition (ET), stable boundary layer (SBL), morning transition (MT)). We expand the idealised ABL simulation to 54 h to also include the second diurnal cycle, investigating the difference of the WT wakes between the diurnal cycle of day 1 and day 2. The ABL BLLAST simulation is now a chapter (chapter 3) of the new version of the manuscript. Chapter 4 is updated with the new idealised ABL simulation. The WT study follows in chapter 5.

[1.2] As described in [1.1], WT modelling is included in the manuscript, leading to a unique and novel approach in the new version of the manuscript. This is stated in the following paragraph of the new version of the manuscript: 'Further, we perform a simulation of an idealised ABL throughout two full diurnal cycles with periodic horizontal boundary conditions. This is for two reasons. First, to investigate the diurnal variation of different atmospheric variables relevant in wind energy research. Second, as real-time atmospheric inflow condition in simulations of a single WT with open horizontal boundary conditions for the investigation of the impact of the CBL, the ET, the SBL and the MT in the course of a day on the wake structure. To our knowledge, this is the first study which investigates the influence of a full diurnal cycle, including the ET and the MT, on the wake of a single WT.'

Referee  Second, the validation with the BLLAST dataset is unconvincing for two reasons. First, the model was run with winds at 10 m s-1 while the observations showed that the wind speed was closer to 3 m s-1. A convincing validation should match wind speeds.[2.1] Second, the model is only validated with three potential temperature profiles and the comparison is largely qualitative. I'd like to see a quantification of the errors as well as validation with additional observations from the BLLAST campaign.[2.2] The discussion of the results then shows that the LES model is able to capture the expected structure of the ABL. The discussion fails to highlight what is novel about these results compared with previous LES studies of the ABL[2.3]

Author  [2.1] We rerun the BLLAST ABL simulation initialised with the measured wind profile from the 0000 UTC radiosonde launch for u and v in addition to $\Theta$ instead of a geostrophic wind of $u = 10$ m s$^{-1}$ and $v = 0$ m s$^{-1}$ and updated the results in chapter 3. In the new version of chapter 3 we investigate the time and space variation of the potential temperature, the individual turbulent kinetic energy (TKE) budget terms integrated over the height of the ABL and as vertical evolution and the temporal evolution of the potential temperature (old Fig. 1). (See Fig. 1 below.)

[2.2] We tried to compare the temporal and spatial evolution of $u$ and $v$ with sodar measurement for $z \leq 800$ m and with UHF wind profiler data for $z > 800$ m. This comparison does not lead to a satisfying agreement after a few hours of simulation. In our opinion, it is very difficult to fit the observed values of the wind speed and the wind direction with the LES. This is for two reasons. First, our numerical simulation strategy. We initialise with the 0000 UTC radiosonde data, however, we do not nudge during the simulation to measured wind profiles. Second,

mesoscale processes. The dynamics of the atmosphere might be dominated by mesoscales processes like mountain-valley flows, which are not considered in the LES. We considered this in the new manuscript as: 'Considering 23 h of simulation with the limited prescribed external forcings, the colder temperature in the lowest levels can be caused by additional large-scale effects, which are not included in our simulation, e.g. colder air close to the surface advecting from the mountains as part of the mountain-plain circulation. '

In addition, we tried to compare the temporal and spatial evolution of $\theta$ with radiometer measurements. Unfortunately, from the 01.07.11 to the 05.07.11 there are not data available.

In summary, we arrive at the conclusion that the validation with the BLLAST data of $\theta$ from the radiosonde launch gives confidence in the ability of the geophysical flow solver EULAG to simulate the diurnal cycle of idealised ABL simulations, which is an important step towards the investigation of the diurnal variation of different atmospheric variables relevant in wind energy research ($u$, $I$) and the investigation of the wake structure in WT simulations. We stated this in the new version of the manuscript as: Introduction: 'We simulate the complete diurnal cycle of one day by using observations from the BLLAST field campaign and the large-eddy simulation (LES) model EULAG. The chosen day was characterized by a surface driven weather situation with minimum larger scale disturbances. Therefore, it is well-suited to validate the geophysical flow solver EULAG.' Conclusion: 'The validation of a full diurnal cycle of the BLLAST field experiment with our LES model EULAG gives confidence in ABL simulations. ' For this purpose, we decide that we do not need a more detail investigation and quantification of the errors for the new main statement of this manuscript, the impact of the diurnal cycle of the atmospheric boundary layer on wind turbine wakes.
[2.3] See detailed answer in [1.2].

Referee  Finally, the grammar, sentence structure and general readability of the manuscript need to be improved.[3.1] I encourage the authors to carefully edit the manuscript and highlight what is unique about their approach/results and resubmit.[3.2]

Author  [3.1] We aim to improve the grammar, sentence structure and general readability of the paper.
[3.2] See detailed answer in [1.2].

**Specific Comments**

Referee: 5  "In this way, this contribution to the special issue of ACP 'The Boundary-Layer Late Afternoon and Sunset Turbulence project' satisfies the purpose of the BLLAST experiment: to provide a dataset for the validation of numerical simulations aiming to study transient BL processes" – This is not necessary to include in the abstract, include major conclusions instead.

Author: We skip this part. Further, we include the major conclusions e.g. in the abstract via: 'A diurnal cycle simulation with the unique dataset gathered during the BLLAST (Boundary Layer Late Afternoon and Sunset Turbulence) field experiment gives confidence in our idealised diurnal cycle simulation of the atmospheric boundary layer performed with the geophysical flow solver EULAG. The diurnal cycle significantly impacts on the wind shear and the atmospheric turbulence. Specifically, a strong vertical wind shear and a change of the wind direction with height occur in the stable boundary layer and during the morning transition, whereas the atmospheric turbulence is much larger in the convective boundary layer and during the evening transition. These different characteristics of the atmospheric boundary layer are well-suited for studying the interaction with a wind turbine wake, by applying real-time turbulent inflow data from the idealised atmospheric boundary layer simulation with periodic horizontal boundary conditions to the wind turbine simulation with open horizontal boundary conditions. The resulting wake is strongly influenced by the stability of the atmosphere and recovers more rapidly under convective conditions during the day, compared to the night. The wake characteristics of the transitional periods are influenced by the flow regime prior to the transition. Further, there is barely seen any difference between the corresponding wake structures throughout two full diurnal cycles.'

Referee: 15 It may not be necessary to include so much detail regarding the ABL in ACP. Is it possible to include the ABL schematic from Stull?

Author: We changed it. Now it is much more compact included in the following sentence: 'Based on thermal stratification and the dominant mechanism of turbulence production/destruction, the ABL is classified into stable, convective and neutral (Stull, 1988).' Further it says: 'These transition periods are referred to as MT and ET and are defined following Grimsdell and Angevine (2002) as the time period in which the sensible heat flux changes sign.'

Referee: 34 "initiated by a positive heat flux" – Use "upward" instead of "positive"

Author: We changed it.

Referee: 73 "the inflow wind field a wind turbine is exposed to strongly influences the wake structure and the turbine loading, both affecting the power production of a wind turbine" – Confusing sentence structure. Consider rewording

Author: We changed it to: 'Atmospheric turbulence has an impact on the power produced by a WT and on the turbine loading (Sathe et al., 2013) because it affects the streamwise extension of the wake, the magnitude of the velocity deficit and the turbulence in the wake. The influence of atmospheric turbulence on these wake characteristics has been investigated in experimental studies considering different atmospheric stratifications (Baker and Walker, 1984; Medici and

Alfredsson, 2006; Chamorro and Porté-Agel, 2010; Zhang et al., 2012, 2013; Tian et al., 2013; Hancock and Pascheke, 2014; Hancock and Zhang, 2015).'

Referee: 83 "Which impact have the individual phases of the diurnal cycle on the physical variables relevant for wind energy applications?" – Confusing, please reword

Author: We changed it to: 'Further, we perform a simulation of an idealised ABL throughout two full diurnal cycles with periodic horizontal boundary conditions. This is for two reasons. First, to investigate the diurnal variation of different atmospheric variables relevant in wind energy research. Second, ...'

Referee: 98 "Most of the performed LES simulations on the characteristics of the BL, mentioned above, prescribe homogeneous surface conditions. However, the Earth's surface is not homogeneous. It is strongly affected by different land use, buildings, and so on. Therefore, considering heterogeneous surface conditions will especially improve the turbulence structure close to the ground" – Without validation or a reference, this is a non-sequitur conclusion. Just because "realistic" surface conditions are implemented, it does not guarantee that the model will more accurately capture the near-surface turbulence structure. Please include a reference to where this has been validated or show this with your data.

Author: We agree with the reviewer, a much more detailed investigation of the simulations with heterogeneous surface would be necessary. We skip this investigation in this manuscript (old chapter 5), because a detailed investigation of the ABL simulation over heterogeneous surface together with the corresponding WT simulations over heterogeneous surface would result in a rather long manuscript. Further, we choose WT simulation throughout two full diurnal cycles as new mayor topic (new chapter 5) of the manuscript, as we thought it fits better to chapter 3 and chapter 4 of the old paper. ABL and WT simulations over heterogeneous surface are considered as a self-consistent topic and future research will include it.

Referee: 258 "validated against the other three measurements" – Name the other three measurements Figure 1 Caption – "The initial starting profile is also plotted for 0000 UTC" – That's the dotted line?

Author: We try to make it more clear by excluding the initial starting profile (dotted line). Fig. 1 (Fig. 3 in new manuscript) now includes the radiosonde measurements (dashed lines) and the LES results (solid lines) for 0 h, 11 h, 17 h and 23 h (other three measurements correspond to the dashed lines at 11 h, 17 h and 23 h). Further, we rename 'measurement' in 'RS' (radiosonde measurement) (definition in the legend of new Fig. 3). See Fig. 1 below.

Referee: 309 "At 2300 UTC, there is a difference prevalent between the measurement and the LES result in the lowest levels." – It'd be nice to see a bit more discussion on this since it's the only simulated profile of the SBL

Author: A detailed discussion of the stable profiles is given in the additional idealised ABL simulation over homogeneous surface considering the SBL, the MT and the low-level jet (LLJ) in chapter 4 of the new manuscript: '... Specifically, in the CBL and during the ET the vertical wind shear is rather small, whereas in the SBL and during MT it is very pronounced. A supergeostrophic situation prevails during the MT near hub height corresponding to a LLJ with a change in wind shear from a positive value below to a negative value above. A supergeostrophic situation also exists in the SBL simulation of Aitken et al., 2014, Fig. 4, Bhaganagar and Debnath, 2014, Fig. 1$a$ and Bhaganagar and Debnath, 2015, Fig. 1. In our simulation, the LLJ is not yet prevalent in the SBL, only a positive wind shear exists between bottom tip and hub height, because the onset time as well as the height of the LLJ depend on the amount of infrared irradiation at night and on the atmospheric situation of the previous day (Bhaganagar and Debnath, 2014, 2015).'

**References**

Aitken, M. L., Kosović, B., Mirocha, J. D., and Lundquist, J. K.: Large eddy simulation of wind turbine wake dynamics in the stable boundary layer using the Weather Research and Forecasting Model, Journal of Renewable and Sustainable Energy, 6, 033 137, doi:10.1063/1.4885111, 2014.

Baker, R. W. and Walker, S. N.: Wake measurements behind a large horizontal axis wind turbine generator, Solar Energy, 33, 5–12, doi:10.1016/0038-092X(84)90110-5, 1984.

Bhaganagar, K. and Debnath, M.: Implications of Stably Stratified Atmospheric Boundary Layer Turbulence on the Near-Wake Structure of Wind Turbines, Energies, 7, 5740–5763, doi:10.3390/en7095740, 2014.

Bhaganagar, K. and Debnath, M.: The effects of mean atmospheric forcings of the stable atmospheric boundary layer on wind turbine wake, Journal of Renewable and Sustainable Energy, 7, 013 124, doi:10.1063/1.4907687, 2015.

Chamorro, L. P. and Porté-Agel, F.: Effects of Thermal Stability and Incoming Boundary-Layer Flow Characteristics on Wind-Turbine Wakes: A Wind-Tunnel Study, Bound-Lay Meteorol, 136, 515–533, doi:10.1007/s10546-010-9512-1, 2010.

Grimsdell, A. W. and Angevine, W. M.: Observations of the afternoon transition of the convective boundary layer, J Appl Meteorol, 41, 3–11, doi:10.1175/1520-0450(2002)041<3C0003:OOTATO>3E2.0.CO;2, 2002.

Hancock, P. and Zhang, S.: A Wind-Tunnel Simulation of the Wake of a Large Wind Turbine in a Weakly Unstable Boundary Layer, Boundary-Layer Meteorology, 156, 395–413, doi:10.1007/s10546-015-0037-5, 2015.

Hancock, P. E. and Pascheke, F.: Wind-tunnel simulation of the wake of a large wind turbine in a stable boundary layer: Part 2, the wake flow, Boundary-layer meteorology, 151, 23–37, doi:10.1007/s10546-013-9887-x, 2014.

Medici, D. and Alfredsson, P. H.: Measurements on a wind turbine wake: 3D effects and bluff body vortex shedding, Wind Energy, 9, 219–236, doi:10.1002/we.156, 2006.

Sathe, A., Mann, J., Barlas, T., Bierbooms, W., and Bussel, G.: Influence of atmospheric stability on wind turbine loads, Wind Energy, 16, 1013–1032, doi:10.1002/we.1528, 2013.

Stull, R. B.: An Introduction of Boundary Layer Meteorology, Dordrecht, Kluwer Academic, 1988.

Tian, W., Ozbay, A., Yuan, W., Sarakar, P., and Hu, H.: An experimental study on the performances of wind turbines over complex terrain, in: 51st AIAA Aerospace Sciences Meeting including the New Horizons Forum and Aerospace Exposition, 07-10 January 2013, Grapevine, Texas, USA, 1-14, doi:10.1115/FEDSM2012-72306, 2013.

Zhang, W., Markfort, C. D., and Porté-Agel, F.: Near-wake flow structure downwind of a wind turbine in a turbulent boundary layer, Exp Fluids, 52, 1219–1235, doi:10.1007/s00348-011-1250-8, 2012.

Zhang, W., Markfort, C. D., and Porté-Agel, F.: Wind-Turbine Wakes in a Convective Boundary Layer: A Wind-Tunnel Study, Bound-Lay Meteorol, 146, 161–179, doi:10.1007/s10546-012-9751-4, 2013.

[Figure]

**Figure 1.** The temporal evolution of the horizontal average of $\Theta(z)$ (solid lines) and the corresponding radiosonde measurements (dashed lines) at 0 h, 11 h, 17 h and 23 h for the lowest 2 km.